# Biodiversity Dynamics in a Ramsar Wetland: Assessing How Climate and Hydrology Shape the Distribution of Dominant Native and Alien Macrophytes

**DOI:** 10.3390/plants14071116

**Published:** 2025-04-03

**Authors:** Fabio A. Labra, Eduardo Jaramillo

**Affiliations:** 1Centro de Investigación e Innovación en Cambio Climático, Facultad de Ciencias, Universidad Santo Tomás, Santiago 8370003, Chile; 2Instituto de Ciencias de la Tierra, Facultad de Ciencias, Universidad Austral de Chile, Valdivia 5090000, Chile

**Keywords:** Ramsar site, macrophyte species distribution modeling, *Elodea densa*, *Egeria densa*, *Schoenoplectus californicus*, precipitation and river flow, invasive alien species (IAS)

## Abstract

Coastal wetlands provide critical ecological services but are threatened by the human, climatic, and hydrological changes impacting these ecosystems. Several key ecosystem services and functions rely on aquatic macrophyte plant species. We integrate 10 years of seasonal monitoring data (2014–2024) and climatic and hydrological datasets to assess how environmental variability influences two dominant aquatic macrophytes—the invasive and non-indigenous *Elodea densa* Planch. Casp. (Hydrocharitaceae) and the native *Schoenoplectus californicus* (C.A.Mey.) Soják—in Chile’s first Ramsar site, Carlos Anwandter, and a Nature Sanctuary. We modeled suitable habitat areas using MaxEnt software with Landsat 8 spectral bands and indices as predictive layers. We found significant recent decreases in temperature, river flow, and water level, with a nonsignificant shift in precipitation. We also observed marked spatial and temporal fluctuations in areas with suitable habitat areas for both macrophytes. Stepwise regression analyses indicated that *Elodea densa* expanded with increasing temperature over time but declined with water level variability. *Schoenoplectus californicus* showed contrasting effects, declining with rising temperature and water levels but expanding with higher precipitation. These findings emphasize the complexity of coastal wetland ecosystems under environmental stress and climate change and the need for further research for the conservation and management of coastal wetlands along migratory flyways such as the Southeastern Pacific Flyway.

## 1. Introduction

Wetlands, particularly coastal wetlands, fulfill numerous ecosystem functions and deliver essential ecosystem services to human society across the world. These include the preservation of native biodiversity, regulation of biogeochemical processes and water cycles, carbon sequestration, provision of a habitat for commercial fisheries, storm surge protection, and opportunities for cultural, recreational, and educational activities [1,2,3,4,5,6,7]. Thus, coastal wetlands are highly-valued ecological zones where terrestrial and marine ecosystems meet and intersect, sharing distinctive features and threats [5,8,9,10]. These ecosystems are often essential refuges for a wide range of flora and migratory coastal birds, as is the case for species that move through the migratory routes of the Pacific coast of South America (Southeastern Pacific Flyway, SEPF) [11,12,13]. With nearly 6500 linear kilometers of continental coastline, the coastal wetlands along continental Chile are thus a crucial element of the SEPF—even with a fragmented distribution that decreases from south to north—since it provides a valuable migratory corridor with high biodiversity, productivity, and endemism [3,4,14].

Despite their importance, coastal wetlands are frequently degraded by a variety of anthropogenic activities including wetland reclamation and land use change, water extraction, and nutrient and pollution overload of incoming waters. These factors often cause cascading effects on abiotic variables and processes, as well as the structure and function of biological hierarchical levels such as primary producers and their consumers (e.g., [15,16,17]). In coastal wetlands, these changes frequently occur in tandem with the effects of global-scale environmental stressors such as the effects of climate change through increasing temperature and sea level rise [5] or the intensity and frequency of storm surges [18]. In Chile, the most vulnerable coastal wetlands are found between 32 °S and 40 °S, and the dominant drivers affecting them are climate change as well as urbanization and land use changes, which affect 41.9% and 52.8% of these wetlands, respectively [7]. Among those, the Rio Cruces Wetland (RCW) near the city of Valdivia (ca. 39.9 °S) stands out, as it harbors Chile’s first Ramsar Site, located in the wetland’s central area. The RCW was formed as a result of large co-seismic continental subsidence following the great Valdivia earthquake of 1960, which caused level decreases of up to 2.5 m [19,20,21]. As a result, land areas that were previously adjacent to the riverine system and occupied by agriculture, stock farming, and marsh forests were transformed into shallow-water or frequently flooded wetland areas [22]. In 1981, the central area of the RCW (Figure 1) was designated as Ramsar Site n°222, given its support of numerous waterbirds, including two endangered species: Coscoroba swan (*Coscoroba coscoroba*) and the White-faced Ibis (*Plegadis chihi*). At that time, it harbored the largest recorded stable nesting population of the Black-necked Swan (*Cygnus melancoryphus*), totaling 3000 individuals. The RCW is modulated by the tidal cycle, as the water level is affected by the inflow of tides through the river mouth at Bahia Corral, with the tidal cycle modulating the water level across the wetland [23]. The RCW currently hosts a high diversity of birds, amphibians, fish, mammals, and aquatic macrophytes [24,25,26].

The RCW has been the subject of several studies following the changes in water quality, after a pulp mill plant (located 15 km upstream of the wetland) began operating and discharging residual waters into the river during February 2004 [27]. The changes in water quality cascaded throughout the widespread stands of the Brazilian elodea, *Elodea densa* (Planch.) Casp. (Hydrocharitaceae) (often reported in the recent literature under the synonym *Egeria densa* Planch. (Hydrocharitaceae)). This invasive alien species was and remains the main food of the black-necked swans at the RCW, and its collapse during 2004 led to massive swan migration and mortality due to emaciation and histopathological liver damage [17,27,28,29,30]. Several studies conducted following the 2004 population decline and subsequent recovery of *C. melancoryphus* have demonstrated the significance of aquatic macrophytes for the RCW ecosystem and its tributary rivers, as many of them can be considered ecosystem engineers [31,32]. Among these, *E. densa* still stands out as a dominant food source for the herbivorous aquatic birds of the wetland, such as black-necked swans and coots, while it also helps to stabilize the benthic sediments in the wetland’s shallow channels and flatlands, playing important roles in the flow of energy and matter, as well as the structure and hydrological dynamics of this ecosystem [28,31,33]. Thus, this invasive alien species (IAS) plays a crucial role in the RCW ecosystem. Along with this IAS, the native macrophytes *Potamogeton lucens* L. (Potamogetonaceae) and *Potamogeton pusillus* L. (Potamogetonaceae) have also been shown to be important trophic items for swans [33] and stabilizers of sub-aquatic sediments [24,34]. In addition to these important food sources for herbivorous birds and ecosystem engineers, most sedimentary tidal flats in the RCW are associated with or surrounded by the California bulrush, *Schoenoplectus californicus* (C.A. Mey.) Sojak (Cyperaceae) (also known as Cattail or Totora in Spanish), which is one of the RCW’s dominant macrophytes and occupies extensive areas along the subaerial ebb tide perimeter [22]. Today, most of these shallow bulrush-occupied tidal flat complexes are mostly found in the southern and middle sectors of the RCW [22]. *Schoenoplectus californicus*, like *E. densa*, stabilizes and models wetland shores or shallow bottoms against external hydraulic stress factors, since its emerged stems or canopy structures capture suspended sediments cf. [35,36,37,38]. In this regard and given the relevance of the effects of climate change on Chile’s coastal wetlands, it is important to determine the potential effects of climatic and hydrological drivers on these aquatic ecosystems.

*Elodea densa* is a submerged perennial monocotyledonous dioecious plant indigenous to Brazil, Uruguay, and Argentina in eastern South America. It is currently recognized as a problematic invasive alien species by the International Union for the Conservation of Nature [39]. Its popularity as an aquarium species has led to the establishment of wild populations in all the continents, particularly across many temperate regions around the world, including Chile, North America, Oceania, Europe, Africa, and Asia [31,39,40,41,42,43,44,45,46,47,48,49]. As mentioned above, it became the main food source for black-necked swans and other herbivorous bird species in the RCW [28,29,30]. It has become an integral part of the food web of this Ramsar Site and adjacent wetland areas, and in so doing stabilizes shallow bottom sediments in the RCW. On the other hand, *S. californicus* is a perennial rhizomatous geophyte. It is a native species widely distributed throughout the Americas and parts of the South Pacific, growing primarily in the subtropical biome [50]. While it is not an important trophic item for herbivorous waterbirds, it is an important ecosystem engineer and provides habitats both for aquatic species as well as for reed birds, herons, and Nutria or Coypu (*Myocastor coypus*). The contrasting life histories and morphologies shown by both species raise the question as to how they may respond to climatic and hydrologic alterations caused by climate change and the implication of these responses for the ecosystem engineering functions provided in the RCW by these dominant aquatic macrophytes.

In this regard, the RCW is subject to disturbance effects resulting from anthropogenic factors, including land use change and climate change, as shown previously by [7]. However, other agents of change include alterations in the Earth’s crust due to seismic cycle dynamics [51] and variability impacts from the warm and cold phases of the El Niño-Southern Oscillation (ENSO) phenomenon, which can lead to water deficits in the contributing basins that supply water flow to coastal wetlands [52,53]. Additional sources of climate variation include the variability linked to the Pacific Decadal Oscillation (PDO), which may result in decadal water deficits, as well as the Atlantic Multidecadal Oscillation (AMO), which similarly influences low-frequency variability [54,55,56]. In addition, the Southern Annular Mode (SAM) or Antarctic Oscillation (AAO) also contributes to large-scale climate phenomena by characterizing the north-south movement of the westerly wind belt encircling Antarctica, a phenomenon which significantly influences climate variability in the middle and high latitudes of the southern hemisphere [52,57,58,59]. Those synergic effects between climate change and large-scale climatic forcing have affected different aspects of the RCW’s climate [52,59] (Figure 2).

Considering the differing morphological strategies and life forms of the two dominant aquatic macrophytes of the RCW, the exotic *E. densa*, and the native bulrush, *S. californicus*, along with their roles as ecosystem engineers in this important Ramsar site and Nature Sanctuary, it is essential to evaluate their responses to climatic and hydrological variations in the wetland area. This study aims to evaluate the relative impact of climatic and hydrological factors on the spatial and temporal distribution patterns of *E. densa* and *S. californicus* over the last decade. We integrate long-term monitoring efforts across a ten-year period (2015–2024), including the systematic occurrence of data collected during the summer seasons integrated with remote sensing images, to fit a species distribution model (SDM) for both species every year. These SDMs allow us to estimate the distribution area occupied by *E. densa* and *S. californicus* within the Ramsar site and to model its response to climatic and hydrological variables over the study area. This will allow us to determine what the relative effects of climatic (temperature and precipitation) and hydrological (flow) drivers are on the spatial and temporal distribution patterns of the dominant aquatic macrophytes in the Rio Cruces Wetland. By examining these species, this research aims to provide insight into how native and exotic aquatic macrophytes respond to climatic and hydrologic forcing, contributing to a broader understanding of ecosystem dynamics in response to environmental change.

## 2. Results

### 2.1. Environmental Variability

The study of the pattern of climatic and hydrologic variability in the study area showed a clear seasonal pattern in all four environmental variables, which, during the autumn and winter months, exhibits decreases in temperature and increases in precipitation, river flow, and water level (Figure 2).

Examination of the general additive models shows that the baseline decades differ significantly from 0 in all four variables (Table 1). The fitted models show significant decreases over the last 10 years (2013 to 2023) in temperature as well as river flow and water level, while precipitation shows a non-significant trend toward earlier rainy seasons (Figure 2, Table 1). The fitted general additive models for temperature, river flow, and water level account for 70–89% of the observed variability, while the seasonal model for precipitation accounts for 55% of the observed variability.

While the observed long-term patterns of change in precipitation for the study area reported previously [52,59] are not reflected in recent changes in the seasonal pattern, the rest of the variables examined do present significant decreases over the recent decade (Figure 2, Table 1). Thus, for these three variables, the models effectively capture the seasonal temperature pattern and the differences between groups. The significant smooth term for months confirms the presence of a strong seasonal trend, while the significant group differences highlight systematic variations between the recent decade versus the baseline. The fact that three out of the four climatic and hydrologic drivers examined have changed relative to their long-term baselines suggests that these drivers and related variables may be affecting the RCW and its surrounding hydrologic basin, with potential effects on the dominant aquatic macrophyte species of the wetland.

### 2.2. Species Distribution Modeling

Despite the high frequency of cloud cover across the study area, it was possible to obtain cloud-free Landsat 8 scenes in all ten spring–summer study seasons (Table 2).

Across all 10 years and for both species, the fitted SDM models showed close fits to the observed presence data, as evidenced by Area Under the Receiver Operating Characteristic Curve (AUC) values above 0.85 in both the training and test cross-validation datasets across all samples for *E. densa* and *S. californicus* (Table 1). This indicates that the models provide very good matches in the classification task, while those years with test AUC values greater than 0.9 present highly reliable fits to the available data [59]. Detailed Receiver Operating Characteristic Curve (ROC) curves for *E. densa* and *S. californicus* are shown in Appendix A, respectively.

The spatial pattern of the fitted MaxEnt habitat suitability maps is shown in Figure 3 and Figure 4. In the case of the exotic macrophyte *E. densa*, we observed a heterogeneous spatial variation in habitat suitability (Figure 3). Thus, patches of high habitat suitability across all the shallow areas of the wetland were observed during the first four years (2015 to 2018). After 2019, habitat suitability decreased in the northern sections of Rio Cruces and northern tributaries, followed by a marked decrease in habitat suitability across the central and southern areas of the wetland (Figure 3e–g). Some of the areas that show decreases in habitat suitability do show signs of increase after 2021, suggesting a fluctuation in the distribution of this species (Figure 3h–j). On the other hand, results for the native bulrush, *S. californicus,* show a patchy distribution across the wetland, associated with shallow tidal flats, with decreases in the values of habitat suitability in the north section of the wetland being observed in the spring–summer seasons of 2017, 2020, 2021, and 2022 (Figure 4a–h). The fluctuation in habitat suitability in the northern sections of the wetland contrasts with the areas in the central and southern parts of the wetland, which tend to present stable patches of the bulrush (Figure 4).

### 2.3. Assessing Interdecadal Variation in the Area of Suitable Habitat

We examined the dynamics of the area of suitable habitat within the RCW. For both *E. densa* and *S. californicus,* we observed that while the area shows yearly oscillations, no significant linear trend is present (Figure 5). Thus, the Ordinary least Squares (OLS) linear regression for *E. densa* was not statistically significant (R^2^ = 0, F(1,8) = 0.005, *p* = 0.943), with the year having no significant effect on the area (β_Year_ = −4 ± 53.8, *p* = 0.943). In the case of *S. californicus*, while a decreasing trend was observed, this was not statistically significant (R^2^ = 0.22, F(1,8) = 2.3, *p* = 0.168), and again, no significant effect of the year on the area was observed (β_Year_ = −87.9 ± 58, *p* = 0.168) (Figure 5). When examining the degree of relative variability, *E. densa* showed a lower Coefficient of Variation (CV) value than *S. californicus* (20.46% and 30.05%, respectively), suggesting that these species show different responses to environmental fluctuations.

After the above-mentioned analyses, we conducted a stepwise linear regression analysis to evaluate the interannual variation in the macrophyte area within the RCW as a function of climatic and hydrological variables. For the IAS *E. densa*, the final model retained three significant predictors: mean annual temperature (T), standard deviation of water level (s.d. Level), and Year. The model intercept was significant, −3733.32 ± 1157.59 (t = −3.23, *p* = 0.018) (Table 3). In this model, the Year had a positive association with macrophyte area (β = 1.83 ± 0.57, t = 3.22, *p* = 0.018), indicating an increasing trend over time. Mean annual temperature (T_Year_) also showed a significant positive effect (β = 6.93 ± 1.72, t = 4.02, *p* = 0.007), suggesting that higher temperatures contribute to an expansion of *E. densa*. In contrast, the standard deviation of water level (s.d. Level) had a significant negative effect (β = −32.94 ± 11.76, t = −2.80, *p* = 0.031), indicating that greater fluctuations in water level may negatively impact the species’ area of suitable habitat.

For the native *S. californicus*, the stepwise linear regression model retained all five predictors: year, mean annual temperature (T_Year_), accumulated annual precipitation (sP_Year_), mean annual water level (Level), and standard deviation of water level (s.d. Level). While the intercept and year were not statistically significant (*p* = 0.093 and *p* = 0.098, respectively), the mean annual temperature (T_Year_) had a significant negative effect on *S. californicus* (β = −6.19 ± 1.85, t = −3.35, *p* = 0.029), suggesting that higher temperatures reduce the area of suitable habitat. Accumulated annual precipitation (sP_Year_) was positively associated with macrophyte area (β = 0.02 ± 0.003, t = 5.13, *p* = 0.007), indicating that increased rainfall favors the expansion of *S. californicus*. However, the mean annual water level (Level_Year_) had a significant negative effect (β = −46.47 ± 9.99, t = −4.65, *p* = 0.010), implying that higher river levels are associated with a reduction in suitable habitat area for *S. californicus*. Lastly, the standard deviation of the water level (s.d. Level) was not found to be a significant predictor (*p* = 0.294) (Table 3).

For *S. californicus*, the selected regression model included non-significant effects (see Table 3). Hence, we also examined a simplified linear model that was fitted that excluded those non-significant variables. This enabled us to compare the stepwise linear regression model to the reduced model, with the Akaike Information Criterion (AIC) determining and selecting the most parsimonious model [60,61]. The reduced model exhibited a higher Akaike Information Criterion (AIC = 55.4) compared to the stepwise model (AIC = 51.7), suggesting that the inclusion of all five predictors provided a better overall model fit while remaining parsimonious (see Appendix A). Despite this, both models identified similar key drivers influencing the *S. californicus* habitat. Mean annual temperature (T_Year_) consistently had a significant negative effect, indicating that higher temperatures reduce suitable habitat areas. Similarly, accumulated annual precipitation (sP_Year_) had a strong positive influence, while mean annual water level (Level_Year_) had a significant negative effect, suggesting that increased water levels reduce the area of suitable habitat. These results confirm that climatic and hydrological variables play a crucial role in determining the extent of the *S. californicus* habitat and that while the reduced model simplifies interpretation, the stepwise model provides a more comprehensive representation of environmental influences.

Overall, the analysis reveals that the IAS *E. densa* responds positively to increasing temperatures and tends to increase over time, while native *S. californicus* is negatively affected by higher temperatures and river levels but benefits from increased precipitation. The significant predictors identified in these stepwise regression models highlight the differential responses of these two macrophyte species to climatic and hydrological variations.

## 3. Discussion

### 3.1. Climatic and Hydrological Variability in the Study Area

Our results show significant seasonal and interannual climatic and hydrological variability within the RCW, with distinct seasonal patterns in air temperature, precipitation, river flow, and water level. While average monthly air temperature, river flow, and water level have all decreased significantly over the last decade (2013–2023), precipitation does not show significant changes, although a trend toward earlier rainy seasons can be appreciated. These findings are consistent with earlier research showing long-term alterations in precipitation patterns and hydrological regimes in the region [52,53,57,58,59]. On the other hand, the observed reduction in air temperature contradicts global and regional warming trends [62], which suggests the possible effects of localized climate influences on the study area, likely resulting from the influence of neighboring oceanic water mass or interactions with meteorological events such as atmospheric rivers [63,64,65]. While recent studies have examined the effects of climate change and hydrological variation on watersheds across Chile (e.g., [66]), the detailed variation in climate and hydrology in some coastal watersheds, such as the section studied of the RCW, has yet to be investigated. Our findings highlight the need for additional research into the impact of climate change on climatic and hydrologic variables in this area of the RCW. This is particularly relevant in light of the recent accreditation of Valdivia as one of Latin America’s first Ramsar wetland cities.

### 3.2. Species Distribution Modeling

Despite the high cloud cover frequency of the study area, we were able to acquire cloud-free Landsat 8 scenes for every year in our study period. The increasing availability of Landsat and Sentinel imagery increased the availability of these spectral satellite imagery, increasing the likelihood of expanding these modeling efforts to the autumn and early spring seasons of the RCW. Thus, our results show how remote sensing and species distribution modeling (SDM) may allow successful monitoring of aquatic macrophytes in similar wetland habitats to those of the RCW [67,68,69].

Across all 10 years analyzed, the fitted species distribution models for both the exotic *E. densa* and the native *S. californicus* demonstrated high predictive performance, with test AUC values exceeding 0.80. Notably, 7 out of 10 *E. densa* MaxEnt models and 8 out of 10 *S. californicus* models achieved test AUC values of 0.90 or higher, confirming their robustness in assessing species–environment relationships and distinguishing between suitable and unsuitable habitats (cf. [70,71,72]). The strong predictive capacity of these remote sensing-based SDMs highlights their utility for understanding macrophyte distribution patterns in the RCW and their potential application in wetland monitoring and conservation planning in other similar ecosystems. While some authors have pointed out some limitations on the use of MaxEnt as an SDM algorithm [73], different large-scale assessments have validated its usefulness and performance in modeling presence-only data with small datasets [71,72,74].

### 3.3. Assessing Interdecadal Variation in the Area of Suitable Habitat

Regarding the interdecadal variation in the area of suitable habitat, we found that *E. densa* experienced a decline in habitat suitability after 2019, followed by some recovery post-2021, while *S. californicus* showed stability in the central and southern wetland but fluctuations in the northern sections. As a result of these changes, no significant linear trend was detected for either species, although both present marked interannual variability, with *E. densa* showing lower relative variability than *S. californicus*. However, stepwise regression analyses revealed that these two species differ in their responses to climatic and hydrological factors. For *E. densa*, higher mean annual temperature emerges as a likely driver of suitable habitat expansion, while greater water level fluctuations negatively impact its distribution. These effects are consistent with previously documented stress response patterns of *E. densa* in Japan [45,46,47]. These studies show that measuring the production of hydrogen peroxide (H_2_O_2_) in plant tissues provides a marker for the stress response. Thus, plants of *E. densa* show reduced growth and a deteriorated physiological condition when tissue H_2_O_2_ concentration exceeds a threshold value [45,46,47]. Field measures across different rivers showed that for temperatures between 10 to 30 °C, the H_2_O_2_ concentration in *E. densa* tissues decreased with increasing temperature [46]. Furthermore, H_2_O_2_ concentration increased with turbulent flow velocity [46]. Increases in water level fluctuations may be associated both with increased flow variability and average value (See Appendix A), as well as with more extreme turbulent flow events. Both conditions could account for the observed negative effect on suitable habit areas at the wetland level. Thus, our observed results for *E. densa* are consistent with available information on the stress drivers for this exotic species. These results and previous observations suggest that other wetlands and continental water areas with low turbulent flow, such as lakes or rivers and channels with low slopes, may have greater habitat suitability for this species. Furthermore, any future changes in water temperature patterns may also alter the habitat suitability for this exotic species. Further studies are needed to determine whether the pattern of H_2_O_2_ concentration previously described in [45,46,47] can also be observed in the RCW wetland or other invaded watercourses in South America.

In contrast with the invasive non-indigenous *E. densa*, the native *S. californicus* exhibited a negative association with temperature and mean annual water level, along with a positive association with accumulated precipitation. This suggests that decreasing temperatures and mean annual water levels may increase suitable habitats for this native species. The observed variation—in both temperature and mean annual water level—suggests that interannual variability in both environmental drivers may account for observed changes in the available area of suitable habitat. Our results are partially consistent with experimental studies conducted in the Sacramento—San Joaquin Bay Delta, California, which show that while transplanted *S. californicus* plants can tolerate more severe frequency, depth, and duration of flooding than other aquatic macrophyte species (*Schoenoplectus acutus* (Muhl. ex Bigelow), Á.Löve and D.Löve (Cyperaceae), and *Typha latifolia* L. (Typhaceae)), this species showed greater vegetation expansion in transplant sites characterized by a deeper surface layer of non-compacted soil in conjunction with shorter durations of flooding [75]. Thus, a decreased mean annual water level may result in shorter durations of flooding in fringe habitats, allowing greater expansion to occur [75]. Furthermore, experimental greenhouse studies of different life history stages of *S. californicus* and *S. acutus* show that longer flooding durations led to lower survival rates for seedlings of both species. These results would be consistent with the conditions of increased mean annual water level, which would entail longer flooding events and a decrease in seedling survival [76]. However, further studies are required that examine the variation in the degree of soil compaction and flooding dynamics (cf. [76]) to determine whether these variables are the proximal cause of observed spatial and temporal variation in the cover and survival of *S. californicus* across the RCW. Our results show that the successful management and conservation of Chile’s first Ramsar site will benefit from the integration of long-term monitoring efforts with remote sensing and modeling strategies to develop useful insights on the potential consequences of ongoing climatic and hydrological changes at the watershed level. In this regard, expanding the temporal and spatial resolution of the remote sensing monitoring program could enhance conservation decision-making, either by complementing the Landsat data with unmanned aerial vehicle (UAV) NDVI measurements of plant condition and status or through the development of spectral phenotyping approaches for both macrophyte species. Given the importance of this Ramsar site, future work should aim toward the integration of a remote sensing monitoring program with both the Ramsar site’s management plan as well as with public awareness and stakeholder engagement.

## 4. Materials and Methods

### 4.1. Study Area and Environmental Variability

The Rio Cruces Wetland (RCW) is situated in the coastal area of the Valdivia River basin and is characterized by a temperate rainy climate, experiencing significant rainfall during the winter months, with limited dry periods and low temperatures in winter [52]. The hydrological dynamics of the RCW sub-basin and its contributing river basins are primarily influenced by precipitation [53,66,77]. The RCW and its surrounding hydrographic basin are situated in a temperate macro bioclimate, significantly influenced by the hyper-oceanic temperate bioclimate and characterized by low thermal oscillation [78]. Remarkably, over 50% of the annual precipitation occurs from May to August [52].

#### 4.1.1. Field Surveys

Starting in 2014, field monitoring was conducted in the Rio Cruces Wetland (RCW) to document the presence of dominant aquatic macrophytes, with a particular focus on the exotic invasive species, the Brazilian Elodea, *E. densa* (Luchecillo in Spanish, [50]) as well on the native California bulrush (also known as Cattail), *S. californicus* (Totora in Spanish, [50]). These are two contrasting species that play important roles in structuring the RCW ecosystem, as they are either an important food source for herbivorous waterbirds (*E. densa*) or play an important role as ecosystem engineers in the RCW (*E. densa* and *S. californicus*). As indicated in the Introduction section, *E. densa* is reported in the recent literature and Chilean collections under the synonym *Egeria densa* Planch. (Hydrocharitaceae), [79,80,81,82]. However, we followed up-to-date nomenclature as indicated in Plants of the World Online [83]. Geo-referenced occurrences of these species were sampled during the austral spring–summer seasons between 2014 and 2015 and 2023 and 2024, yielding a total of 10 years. For each of these years, we located and recorded large patches of these two species in the RCW, selecting patches with a diameter equal to or greater than 30 m, which corresponds to the spatial resolution of Landsat scenes. For each of these stands, the geographic coordinates of the center of the stand were registered using a Global Positioning System navigator (GPS), with a WSG84 coordinate system with a UTM datum (18S zone). The presence data for large monodominant patches were then used to fit species distribution models for each species across the wetland in each of these 10 spring–summer seasons.

Previous studies have extensively documented the presence of both species in the RCW [24,30,31,34,50], including the work by [50], which serves as a field guide to vascular aquatic plants in Chile. Representative specimens of *E. densa* and *S. californicus* from the RCW are available at the Herbarium of Concepción (CONC) [79], as cited by [82]. Full details are provided in the Appendix A.

#### 4.1.2. Environmental Variability

To describe the climatic variability in the Rio Cruces Wetland (RCW), we obtained and analyzed hourly time series records of air temperature (T_hour_, measured in °C) and precipitation (P_hour_, measured in mm) from the Pichoy Airport meteorological station (39.65667° S, 73.08722° W), located 22 km northeast of the city of Valdivia (Figure 1). To determine the hydrologic variability in the area in which the RCW is located, we utilized daily time series of river flow (m^3^/s) and water level (m) data from the Rucaco hydrological station (39.55° S, 72.90° W), located 42 km northeast of Valdivia, in the northern upstream reaches of the Rio Cruces River (Figure 1). Air temperature and precipitation records at Pichoy Airport are available from 1 January 1966, while daily river flow and water level at Rucaco hydrologic station are available from 1 May 1969 and 1 January 2000, respectively. Time series were examined using records up to 31 December 2023. Each hourly time series (T and P) was inspected to identify regions with missing values, which were then interpolated by fitting a structural time series model to capture observable dynamics and impute missing data. Missing value imputation was carried out using R’s imputed TS library’s na_kalman function, which was used for this approach [84,85]. This involves applying a Kalman filter to represent the time series and interpolating missing observations. The Kalman filter is a technique for estimating the dynamics of a linear dynamical system using partial observations with white additive noise or equal variance at all frequencies. The former is useful when noise or inaccuracy is detected in measurements and an accurate estimate of the system’s dynamics is required [86]. Once the imputed missing values were introduced in the hourly T and P time series, these were summarized at a daily level, calculating mean daily values. These daily time series for T, P, Flow, and Level were then used to (i) assess recent interdecadal variation relative to the long-term available baseline and (ii) assess recent patterns of variability of climatic and hydrological drivers across the studied ten-year period.

#### 4.1.3. Assessing Interdecadal Variation

To assess the interdecadal variation, the daily time series for all four variables (T, P, Flow, and Level) were summarized as monthly average values. These monthly time series were then modeled by using the Generalized Additive Model (GAM) framework with cyclic cubic splines, which are useful for periodic data (e.g., months in a year, hours in a day). This ensures that the start (January) and end (December) of the cycle match smoothly, describing any phenological or seasonal cycles present. We included a Group linear predictor, which identified data corresponding to the baseline and data in the most recent decade (2013–2023). The baseline was defined by considering the starting date for each time series. All analytic procedures were carried out using R, with mgcv and ggplot libraries [85,87,88,89,90,91].

#### 4.1.4. Assessing Recent Climatic and Hydrological Variability

To examine recent climatic and hydrological variability in climatic and hydrological drivers that may affect wetland structure and dynamics, we studied the monthly time series and calculated a set of potentially relevant variables. Given that the field sampling was conducted during the late spring or summer austral seasons (see Table 1), we focused on data for the years prior to the studied sampling dates. Thus, climatic and hydrologic variables were calculated for the years 2014 to 2023. Based on the monthly time series, we calculated mean annual values for all four variables (T, P, Flow, and Level), as well as annual standard deviation values for P, Flow, and Level. In addition, we also determined accumulated annual precipitation values as well as the year as a time variable. The resulting nine annual time series were then examined to exclude any variables with a high degree of multicollinearity, measured as the variance inflation factor [92,93], by using a threshold value of 10, as commonly used in the literature. This procedure was implemented using the vifstep function in the R library usdm [93]. As a result, five variables were retained: year (VIF = 6.05), mean annual Temperature (VIF = 5.77), mean annual water level (VIF = 3.64), annual standard deviation of water level (VIF = 2.30), and cumulative annual precipitation (VIF = 5.18). All analytic procedures were carried out using R [85].

### 4.2. Remote Sensing Image Acquisition and Processing

To fit SDMs for each species in each of the ten austral spring–summer sampling seasons, we used remote sensing layers extracted from a Landsat 8 Operational Land Imager (OLI) scene. In each spring–summer sampling season, we downloaded the closest available OLI scene located in path/row combination 233/88 of the Worldwide Reference System 2 (WRS-2). This scene encompasses the entire RCW and is centered at 40°19′20” S, 72°51′00” W. We processed bands 2–7 of the OLI scene to generate a set of SDM predictor Geographic Information System (GIS) layers, following previous studies in this wetland [17,28,29]. Briefly, bands were radiometrically calibrated using Landsat 8 radiance rescaling factors provided in the Landsat image metadata file, and top-of-atmosphere spectral radiance values (L_λ_, W·(m^2^·sr·μ m)^−1^) were calculated. These L_λ_ values were then converted to top-of-atmosphere reflectance percentages (R_TOA_) [94], and the atmospheric correction for Case-2 turbid waters was applied using the path extraction method [17,95]. Following these corrections, the bands were clipped to the study area, and two spectral indices were calculated for each scene: a chlorophyll proxy (CHL = Blue/Green = Band 2/Band 3) [17] and the normalized difference vegetation index (NDVI = (NIR − Red)/(NIR + Red) = (Band5 − Band4)/(Band5 + Band4), where NIR corresponds to the Near Infrared band [96]. To limit the modeling to wetland areas and river courses, a river raster layer that masks terrestrial pixels was generated using the existing cartography of watercourses in the Region, as well as previous raster layers generated by previous Landsat-based studies [97,98,99]. This resulted in a set of nine predictive GIS layers with a spatial resolution of 30 m pixel size that characterize the studied scenes. For each of the ten years studied, all nine GIS layers were used to fit an SDM that provides a spatially explicit estimate of macrophyte habitat suitability distribution, as described in the following section. All GIS layer processing was carried out using GIS, and statistical data processing was conducted utilizing the dplyr, terra, and ggplot2 packages within the R statistical computing environment [91,100,101] as well as QGIS version 3.20.3 Odense [102].

### 4.3. Species Distribution Modeling

SDMs were fitted using Maximum Entropy species distribution modeling software (MaxEnt v.3.3). MaxEnt uses information on spatial occurrences or presences and GIS layers or features to estimate the presence probability function across the study area [71,72,74,103,104,105,106]. Comparative studies have shown that MaxEnt performs better in relation to other methods developed to analyze presence-only datasets (e.g., [72,74,107], even in situations where the sample size is small [108,109,110]. The resulting maximum entropy model has been shown to be equivalent to maximizing the likelihood function of a spatial inhomogeneous Poisson point process [111,112,113], and as a result, its output can be interpreted as providing relative density estimates across space [111,113]. Hence, the resulting probability measure provides a continuous estimate of probability, which measures environmental habitat suitability (HS) for the species under study. MaxEnt SDMs for each of these two macrophytes in the 10 years studied were fitted using a 5-fold cross-validation scheme, thus allowing every occurrence data point to be used as part of the training and evaluation datasets [71,72,74,103,104,105]. To measure how well each model discriminates presences more accurately than a random prediction, we used the area under the curve (AUC) statistic for the Receiver Operating Characteristic (ROC), with values greater than or equal to 0.8 being considered indicative of a very good or excellent fit, while values greater than 0.9 are considered as evidence of an outstanding or highly reliable fit to the available data [60]. Fitted models were later projected over the Rio Cruces Wetland, using the same GIS predictive layers, with the resulting species distribution HS map allowing us to identify where suitable environments for each of the aquatic macrophytes are found in the wetland.

### 4.4. Statistical Analysis

The overall variation in distribution among the two study species was summarized by analyzing the spatially explicit estimation of probability (or HSI values) across the wetland. First, we generated visual representations of the time series of probability maps for each aquatic macrophyte species. Second, we estimated the suitable area available each year for both the exotic *E. densa* and the native *S. californicus*. To do this, each annual probability map for each species was converted into a presence-absence map by applying a threshold value τ to the predicted presence probabilities of each species. This threshold was chosen by using a probability threshold value that maximizes the sum of sensitivity and specificity (MSS) [114]. The resulting time series of presence-absence maps delineates the estimated distribution dynamics for both macrophytes. We then estimated the area of available suitable habitats for each species over the study period by calculating the number of occupied pixels in each year and subsequently multiplying this figure by the surface area of a pixel in a Landsat image (900 m^2^). This yielded a time series of estimated areas of suitable habitat for each species.

To quantify the degree of interannual variability in the area of suitable macrophyte habitat over time within the RCW, we calculated the Coefficient of Variation (CV) and defined the ratio of the standard deviation to the mean, expressed as a percentage. This allowed us to assess the relative fluctuations in habitat availability across the two macrophytes studied [115]. A low CV indicated stability in suitable habitat areas, suggesting consistent environmental conditions, whereas a high CV reflected significant temporal variation, potentially driven by climate variability, hydrological cycle changes, or other ecological disturbances. This approach allowed for a standardized comparison of habitat variability across different regions and time periods. To assess the relative effect of climatic and hydrological variables, we then conducted a stepwise linear regression analysis separately for *E. densa* and *S. californicus*. In each species’ regression model, the response variable was the annual area of suitable habitat, while the predictor variables were selected after removing any collinear variables (see Section 4.1.4). The selected variables were year, mean annual temperature (T_Year_, °C), accumulated annual precipitation (sP_Year_, mm), mean annual water level (Level_Year_, m), and annual water level standard deviation (s.d. Level, m). Stepwise regression was implemented using the function stepAIC from R’s MASS library [116]. This function allows us to iteratively select the most relevant predictors based on statistical significance, optimizing model fit while minimizing multicollinearity. We then examined the plots of observed and predicted annual areas of suitable habitats, comparing them with the linear model prediction based on Year alone. In those cases where the selected regression model included non-significant effects, a reduced linear model was fit, excluding non-significant variables. This allowed us to compare the stepwise linear regression model with the reduced model, using the Akaike Information Criterion (AIC) to identify and select the most parsimonious model [61,117].

## 5. Conclusions

Our study highlights the complex interactions between climatic and hydrological variability and the distribution of native and invasive macrophytes in Chile’s first Ramsar site, the RCW wetland. Over the past decade, we observed significant decreases in temperature, river flow, and water levels, alongside fluctuating patterns in suitable habitats for *E. densa* and *S. californicus*. Our findings indicate that *E. densa* expands with increasing temperatures but declines with greater water level fluctuations, while *S. californicus* responds positively to precipitation and negatively to temperature and water level increases. These contrasting responses underscore the need for targeted monitoring and management strategies that allow the study of the impacts of ongoing environmental changes in the RCW. The integration of long-term monitoring efforts, remote sensing-based species distribution modeling, and adaptive management approaches will be key to ensuring the conservation and resilience of this critical coastal wetland ecosystem. Further research on physiological stress indicators, hydrological restoration, and climate change adaptation will enhance our ability to manage and protect the RCW wetland in the face of future environmental challenges.

## Figures and Tables

**Figure 1 plants-14-01116-f001:**
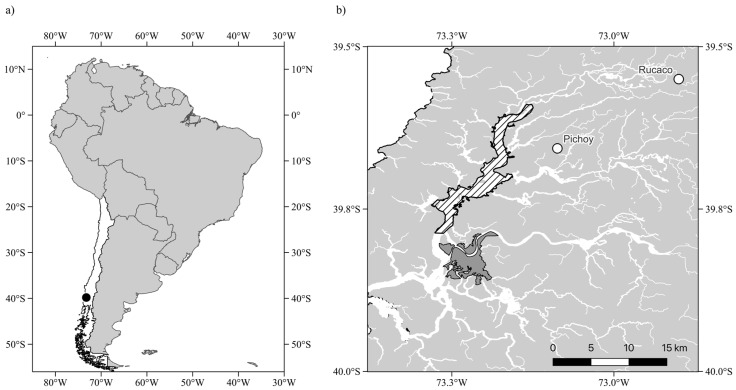
Location of the Rio Cruces Wetland (RCW) in southern Chile. The Figure shows (**a**) South America, with continental Chile highlighted in white, and other countries shaded in grey. The location of the RCW is shown by the filled black circle. (**b**) The location of the central area of the wetland. The hatched polygon highlights the location of the Monumento Nacional Santuario de la Naturaleza Río Cruces y Chorocamayo, Sitio Ramsar Carlos Anwandter, located north of the city of Valdivia, which is indicated by the dark gray polygon. Open circles show the location of the weather station at Pichoy Airport (39.65667° S, 73.08722° W) and the Rucaco hydrological station (39.55 °S, 72.90 °W) in the upstream sector of the Rio Cruces River.

**Figure 2 plants-14-01116-f002:**
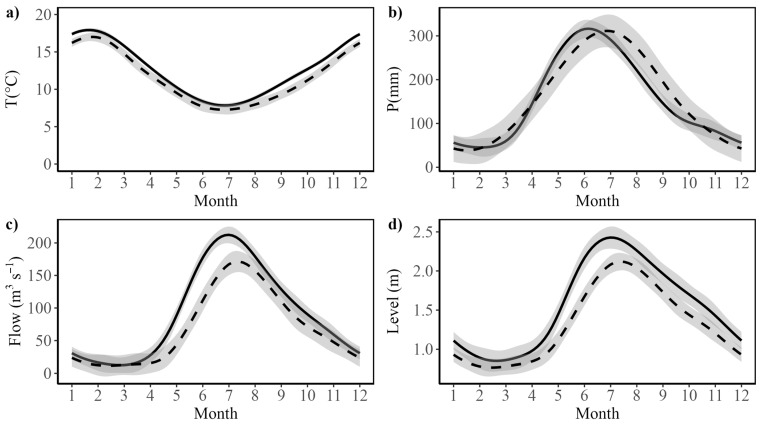
Assessment of changes in long-term patterns in seasonal fluctuations in climatic and hydrologic drivers affecting the RCW. (**a**) Average monthly temperature, T (°C), (**b**) monthly cumulative precipitation, P (mm/month), (**c**) average monthly river flow, Flow (m^3^/s), and (**d**) average monthly water level, Level (m). The continuous black lines show the fit of a cyclic general additive model (GAM) to the seasonal variation across the annual cycle. Data in (**a**,**b**) correspond to the weather station at Pichoy Airport (see Figure 1). Data shown in (**c**,**d**) correspond to variables measured at Rucaco hydrological station (see Figure 1). The observed variation for the reference baseline period (see Table 1) is represented by black continuous lines, whereas the dashed line indicates the variation observed during the recent decade (2013–2023) and the grey shaded bands show the corresponding 95% confidence intervals for the fitted GAMs.

**Figure 3 plants-14-01116-f003:**
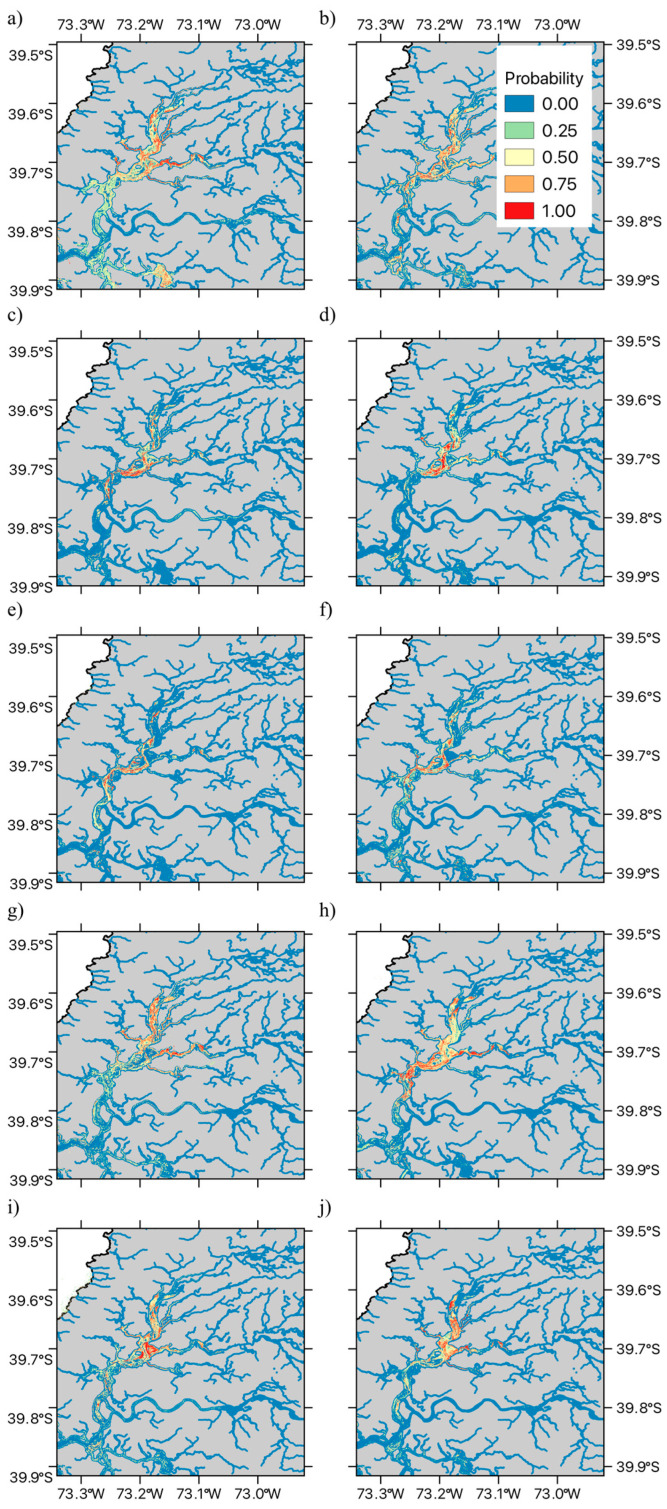
Spatial time series of SDMs for *Elodea densa* (Planch.) Casp. (Hydrocharitaceae). The Figure illustrates the data for austral spring–summer seasons in (**a**) 2015, (**b**) 2016, (**c**) 2017, (**d**) 2018, (**e**) 2019, (**f**) 2020, (**g**) 2021, (**h**) 2022, (**i**) 2023, and (**j**) 2024. The Figure shows the estimated presence probability of *E. densa* in each of the modeled pixels, with the color palette ranging from light blue (zero) to red (one), indicating increasingly greater probability of observing *E. densa* (i.e., the estimated habitat suitability index).

**Figure 4 plants-14-01116-f004:**
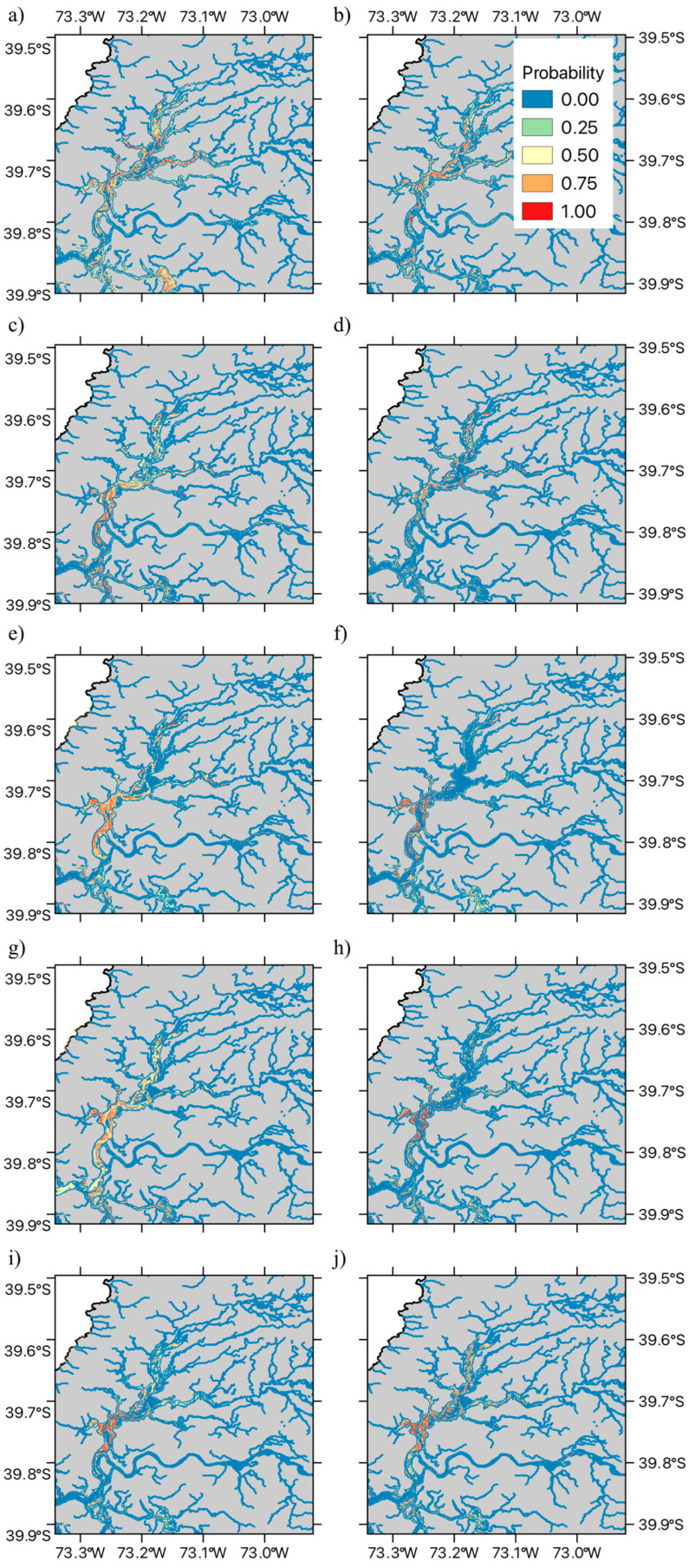
Spatial time series of SDMs for *Schoenoplectus californicus* (C.A. Mey.) Sojak (Cyperaceae). The Figure illustrates the data for the austral spring–summer seasons in (**a**) 2015, (**b**) 2016, (**c**) 2017, (**d**) 2018, (**e**) 2019, (**f**) 2020, (**g**) 2021, (**h**) 2022, (**i**) 2022, and (**j**) 2023. The Figure shows the estimated presence probability of *S. californicus* in each of the modelled pixels, with the color palette ranging from light blue (zero) to red (one), indicating an increasingly greater probability of observing *S. californicus* (i.e., the estimated habitat suitability index).

**Figure 5 plants-14-01116-f005:**
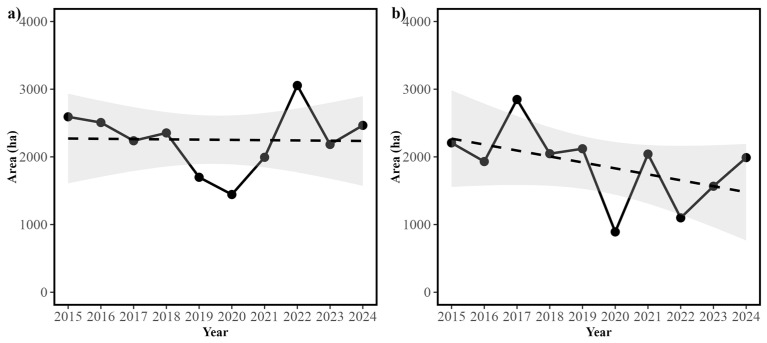
Temporal variation in the distribution area of the two dominant aquatic macrophyte species within the RCW across the ten years studied. This Figure shows (**a**) temporal variation in the area of suitable habitat (measured in km^2^) for *Elodea densa* (Planch.) Casp. (Hydrocharitaceae) and (**b**) temporal variation in the area of suitable habitat (measured in km^2^) for *Schoenoplectus californicus* (C.A. Mey.) Sojak (Cyperaceae). Filled circles and black solid lines show the observed Areas for both species. Dashed lines in figures (**a**,**b**) indicate the fitted linear trend, with their confidence intervals shown in grey.

**Table 1 plants-14-01116-t001:** General additive model fit to describe the interdecadal variation in seasonal variability in climatic and hydrological drivers at the RCW. The Table shows the average monthly air temperature and precipitation (T (°C) and P (mm), respectively) at Pichoy Airport, as well as the average monthly river flow and water level (Flow (m^3^/s) and Level (m), respectively) at Rucaco hydrological station. The table shows the estimated parameter values and standard error, as well as the level of significance. The additive effects of the term associated with the intercept (corresponding to the effect of the baseline decades) as well as of the 2013–2023 decade are estimated. The term s(Group) reflects the degree of fit of the nonlinear function adjusted to the seasonal variation over the 12 months of the year. The F statistic and effective degrees of freedom (df) are reported. The level of significance in each estimate is represented according to the following symbols: ns: *p* ≥ 0.05; ***: *p* < 0.001.

Variable	T (°C)	P (mm)	Flow (m^3^s^−1^)	Level (m)
Intercept	13.00 ± 0.06 ***	153.06 ± 7.40 ***	88.25 ± 2.11 ***	1.53 ± 0.02 ***
2013–2023	−1.01 ± 0.13 ***	4.36 ± 8.31 ^ns^	−21.69 ± 4.64 ***	−0.23 ± 0.03 ***
s(Group)	6.85	5.40	6.292	6.415
F(df)	647.7 (8) ***	100.1 (8) ^ns^	149.9 (8) ***	114.5 (8) ***
GCV	1.69	7425.90	2178.90	0.070294
R^2^_adj_	0.89	0.55	0.70	0.80

**Table 2 plants-14-01116-t002:** Summary of the Landsat 8 scenes and Maximum Entropy (MaxEnt) SDM statistics for the two dominant aquatic macrophytes at the RCW across the ten years studied. The table presents summary information for (a) *Elodea densa* (Planch.) Casp. (Hydrocharitaceae) and (b) *Schoenoplectus californicus* (C.A. Mey.) Sojak (Cyperaceae). The table shows that for each of the ten years, the list of Landsat 8 images was analyzed to model the distribution of these aquatic macrophytes in the study area, indicating the date of the scenes. The table also shows the number of monodominant macrophyte patches with a diameter > 30 m (N), as well as the average Area Under the Receiver Operating Characteristic Curve (AUC) value ± 1 standard error for the training and test cross-validation sets, as well as the average Maximum test sensitivity plus specificity Cloglog threshold (MSS). This threshold allows the estimation of the occupied vs. unoccupied areas of the wetland, by filtering the projected probability maps.

Year	L8 Scene Date ^1^	N	AUC _Train_	AUC _Test_	MSS
(a) *Elodea densa*					
2015	28/01/2015	26	0.93 ± 0.005	0.89 ± 0.027	0.36 ± 0.098
2016	30/12/2015	353	0.93 ± 0.001	0.92 ± 0.003	0.35 ± 0.02
2017	30/11/2016	46	0.97 ± 0.001	0.95 ± 0.008	0.19 ± 0.029
2018	05/02/2018	72	0.95 ± 0.002	0.94 ± 0.012	0.19 ± 0.03
2019	14/01/2019	94	0.97 ± 0.001	0.97 ± 0.003	0.23 ± 0.063
2020	11/02/2020	37	0.96 ± 0.001	0.93 ± 0.015	0.37 ± 0.071
2021	08/03/2021	352	0.89 ± 0.001	0.88 ± 0.005	0.38 ± 0.038
2022	21/12/2021	64	0.9 ± 0.002	0.87 ± 0.016	0.3 ± 0.058
2023	03/02/2023	37	0.93 ± 0.003	0.9 ± 0.015	0.35 ± 0.022
2024	21/01/2024	69	0.95 ± 0.001	0.93 ± 0.012	0.21 ± 0.034
(b) *Schoenoplectus californicus*					
2015	28/01/2015	28	0.92 ± 0.002	0.89 ± 0.007	0.23 ± 0.068
2016	30/12/2015	204	0.93 ± 0.002	0.92 ± 0.007	0.3 ± 0.059
2017	30/11/2016	18	0.95 ± 0.003	0.93 ± 0.017	0.46 ± 0.11
2018	05/02/2018	38	0.96 ± 0.001	0.94 ± 0.009	0.24 ± 0.1
2019	14/01/2019	40	0.95 ± 0.001	0.93 ± 0.008	0.35 ± 0.035
2020	11/02/2020	50	0.96 ± 0.001	0.95 ± 0.004	0.45 ± 0.099
2021	08/03/2021	132	0.9 ± 0.002	0.89 ± 0.006	0.38 ± 0.068
2022	21/12/2021	29	0.96 ± 0.003	0.94 ± 0.012	0.44 ± 0.127
2023	03/02/2023	309	0.92 ± 0.001	0.91 ± 0.005	0.31 ± 0.029
2024	21/01/2024	300	0.94 ± 0.001	0.93 ± 0.005	0.28 ± 0.07

^1^ Scene acquisition dates are shown in format dd/mm/YYYY.

**Table 3 plants-14-01116-t003:** Results of the stepwise linear regression model evaluating the interannual variation in macrophyte area within the RCW as a function of climatic and hydrological variables. The table shows results for (a) *Elodea densa* (Planch.) Casp. (Hydrocharitaceae) and (b) *Schoenoplectus californicus* (C.A. Mey.) Sojak (Cyperaceae). The selected predictor variables considered include mean annual temperature (T_Year_ (°C), accumulated annual precipitation (sP_Year_ (mm)), mean annual water level ((Level_Year_ (m)), and annual water level standard deviation (s.d. Level (m)). The table shows fitted regression coefficients with their standard errors (β ± SE), t-Student values, and *p*-values for each predictor retained in the final model. Model fit statistics, including the adjusted R^2^ and overall model significance, are also reported. The level of significance in each estimate is represented according to the following symbols: ns: *p* ≥ 0.05; **: *p* < 0.01; ***: *p* < 0.001. bold: significant *p* values.

Species and Variables	β ± SE	t	*p*-Value
(a) *Elodea densa*			
Intercept	−3733.32 ± 1157.59	−3.23	**0.018 *****
Year	1.83 ± 0.57	3.22	**0.018 *****
T_Year_ (°C)	6.93 ± 1.72	4.02	**0.007 *****
s.d. Level (m)	−32.94 ± 11.76	−2.80	**0.031 ****
(b) *Schoenoplectus californicus*			
Intercept	3072 ± 1395	2.20	0.093 ^ns^
Year	−1.46 ± 0.68	−2.15	0.098 ^ns^
T_Year_ (°C)	−6.19 ± 1.85	−3.35	**0.029 ****
sP_Year_ (mm)	0.02 ± 0.003	5.13	**0.007 *****
Level (m)	−46.47 ± 9.99	−4.65	**0.010 ****
s.d. Level (m)	14.95 ± 12.38	1.20	0.294 ^ns^

## Data Availability

The raw data supporting the conclusions of this article will be made available by the authors upon request.

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
