# Peer review of "Biodiversity Dynamics in a Ramsar Wetland: Assessing How Climate and Hydrology Shape the Distribution of Dominant Native and Alien Macrophytes"

_plants, 2025, doi:10.3390/plants14071116_

Round 1
Reviewer 1 Report
Comments and Suggestions for Authors
Biodiversity Dynamics in a Ramsar Wetland: Assessing how Climate and Hydrology Shape Distribution of Dominant Native and Alien Macrophytes
This is a well-designed study that presents original information, as well as analyzing data from others, about the Ramsar Wetland in Chile. It provides a thorough review of the site and the primary research and published studies relevant to it. This study will be a paper that is well-cited in the future as a current and forward-looking portrayal of the area. The tables and figures are appropriate and clear. There are some wording and grammatical changes I suggest for consideration (highlighted in red throughout the narrative), and numerous run-on sentences should be broken up for clarity and to make the paper maximally readable. I recommend accepting the paper, and with the incorporation of minor changes in its revision, the paper should be ready for publication. This is clearly a baseline study and reference work for the RCW, and I am pleased that the authors did such a thorough endeavor with it.
Here are comments regarding specific areas within the paper, recommended wording changes, and some questions I inserted into the manuscript. These are indicated by highlighting in red throughout the narrative.
It would be useful to cite the Herbarium or Herbaria which curate collections of RCW plants, including E. densa and S. californicus. If the authors of this paper collected any representative specimens and plan to place them in an Herbarium, that would be helpful to mention. If there have been exsiccata or other representative collections of plants that include these species from the RCW, it would be useful to cite the collection and list the link at which digital images can be viewed (including phenological scoring, if available). It might be that there are RCW plants curated at PUCV – and I’m sure that by contacting any of the authors of the paper cited below, or checking the electronic databases, that could be determined. Here is a paper that be helpful in this regard. Cordero S.; López-Aliste M.; Gálvez F.; Fontúrbel F.E. Herbarium collection of the Pontificia Universidad Católica de Valparaíso (PUCV), Chile. Biodiversity Data Journal 2022, 10, e90591. https://doi.org/10.3897/BDJ.10.e90591
In the References section, some years of publication are in bold, while others are not (2025 and 2025), and that should be corrected, following the preference/directions of the journal. The References section should be proofread to make certain that all citations are consistent in their presentations of the literature included.
In the References bibliography there are several places in which E. densa and S. californicus are not italicized, and I have indicated these in red highlight. I did not check to make certain that all references are cited in the narrative or that all doi links are functional, but that should be verified before publication.
Also in the References list, see reference number 87 in the manuscript, lines 811 – 813, are all of the authors listed? After the last author appears, there is “et al”. If there are additional authors, they should be included. Dormann, C.F.; Elith, J.; Bacher, S.; Buchmann, C.; Carl, G.; Carré, G.; Marquéz, J.R.G.; Gruber, B.; Lafourcade, B.; Leitão, P.J.; et al.???? Collinearity: A Review of Methods to Deal with It and a Simulation Study Evaluating Their Performance. Ecography 2013, 36, 27–46, doi:10.1111/j.1600-0587.2012.07348.x.
Schoenoplectus californicus is referred to in the text as “cattails”, and I recommend changing those references to “California bulrush” – which is what it is known as nearly universally (you can see the “common names” in any of the links below). I know it is awkward to call a species native to Chile and California – the “California” bulrush – and maybe you could use that once for just the first reference to its common name, then just call it “bulrush” after that. “Tule” is another informal generic name for Schoenoplectus species, but I think bulrush is a better common name.
The name Egeria densa Planch.is viewed by some as being a synonym of Elodea densa (Planch.) Casp. I don’t know if it is necessary to mention this or not, but Egeria is certainly in common usage, though Kew recognizes Elodea as its accepted name. See the links that follow (these don’t all need to be cited). It would be helpful to cite at least a couple of sites that have good morphological/distributional descriptions – and these also have links to other approaches/studies of the taxa as well (ecological studies, etc.).
World Flora Online https://about.worldfloraonline.org/consortium-members/royal-botanic-gardens-kew
https://wfoplantlist.org/taxon/wfo-0000770746-2024-12?matched_id=wfo-0000770732&page=1 Elodea densa (Planch.) Casp.
Kew Royal Botanic Gardens. Plants of the World Online. Schoenoplectus californicus (C.A. Mey.) Sojak https://powo.science.kew.org/taxon/urn:lsid:ipni.org:names:229810-2
https://www.worldfloraonline.org
North Carolina Extension Gardener Plant Toolbox. https://plants.ces.ncsu.edu/plants/elodea-densa/common-name/brazilian-waterweed/
Jepson eFlora: Taxon Page for Egeria densa https://ucjeps.berkeley.edu/eflora/eflora_display.php?tid=23849
Jepson eFlora: Taxon Page for Schoenoplectus californicus https://ucjeps.berkeley.edu/eflora/eflora_display.php?tid=43573
Here are some United States federal sites:
United States Department of Agriculture Plants Database. Plant List of Attributes, Names, Taxonomy and Symbols https://plants.usda.gov/ Egeria densa (Brazilian waterweed) https://plants.usda.gov/plant-profile/EGDE; Schoenoplectus californicus https://plants.usda.gov/plant-profile/SCCA11
United States Department of Agriculture National Invasive Species Center. Brazilian waterweed, Egeria densa https://search.invasivespeciesinfo.gov/search?query=Egeria+densa&affiliate=nisic&op=Submit+Search
Brazilian Waterweed | National Invasive Species Information Center
https://www.invasivespeciesinfo.gov/aquatic/plants/brazilian-waterweed Brazilian Waterweed Scientific Name Egeria densa Planch. (ITIS) Common Name Brazilian...Plants of California's Wildlands - Egeria densa California ...
Aquatic Plants | National Invasive Species Information Center
https://www.invasivespeciesinfo.gov/aquatic/plants Griseb. (ITIS) Brazilian Waterweed Egeria densa Planch. (ITIS) Caulerpa, Mediterranean
Invasive Species Profiles List | National Invasive Species Information Center
https://www.invasivespeciesinfo.gov/species-profiles-list Raddi (ITIS) Brazilian Waterweed Egeria densa Planch. (ITIS) British Yellowhead
Overall, the paper is extremely useful and will be a foundational study for the site. I think that after revision it will be an excellent publication, and I look forward to seeing it in its final form.

Author Response
Please see the attachment for a detailed response to the reviewer's comments

Reviewer 2 Report
Comments and Suggestions for Authors
In the manuscript Biodiversity Dynamics in a Ramsar Wetland: Assessing how Climate and Hydrology Shape Distribution of Dominant Native and Alien Macrophytes “authors Fabio A. Labra and Eduardo Jaramillo aimed to provide insight into how native and exotic aquatic macrophytes respond to climatic and hydrologic forcing, contributing to a broader understanding of ecosystem dynamics in response to environmental change.
Abstract
L 16 To do that, we modelled suitable habitat area using… Delete “To do that”
The abstract can serve as a stand-alone document, which succinctly describes both potential use and conclusions.
Key words: Ok
Introduction
L 57 in the wetland's central area…. Parenthesis is missing.
L 62 s [22] (Manzano-Castillo et al. 2020)…. Why the number AND the name of the author?
L 82] . Delete space
L 90 [31,32] . Delete space
The introduction is informative, precise, and comprised of relevant content. The literary structure of the introduction is good, containing key information about the work and on the problem under study.
Materials and methods
Data analysis are clearly explained in the methods section and it is well structured.
Results
The results are well presented, figures and tables are correct.
Discussion
It would be good if the authors measured the H2O2 concentration in Egeria densa tissues.
It lacks concrete proposals for managing RCW and brief conclusions of the MS.
General comments
Punctuation marks are not always in the right places.
The paper has an appropriate structure. In general, I find this work interesting.
My suggestion: minor revision
Author Response

(The authors gave the same response as above.)
